# Controlled bio-inspired self-organised criticality

**Tjeerd V. olde Scheper** *

School of Engineering, Computing and Mathematics, Oxford Brookes University, Wheatley Campus, Oxford, United Kingdom

* tvolde-scheper@brookes.ac.uk

**Data Availability Statement:** The data are available at Zenodo: Code: 10.5281/zenodo.5726935 Data: 10.5281/zenodo.5727044.

**Funding:** The author(s) received no specific funding for this work.

## Abstract

Complex biological systems are considered to be controlled using feedback mechanisms. Reduced systems modelling has been effective to describe these mechanisms, but this approach does not sufficiently encompass the required complexity that is needed to understand how localised control in a biological system can provide global stable states. Self-Organised Criticality (SOC) is a characteristic property of locally interacting physical systems, which readily emerges from changes to its dynamic state due to small nonlinear perturbations. These small changes in the local states, or in local interactions, can greatly affect the total system state of critical systems. It has long been conjectured that SOC is cardinal to biological systems, that show similar critical dynamics, and also may exhibit near power-law relations. Rate Control of Chaos (RCC) provides a suitable robust mechanism to generate SOC systems, which operates at the edge of chaos. The bio-inspired RCC method requires only local instantaneous knowledge of some of the variables of the system, and is capable of adapting to local perturbations. Importantly, connected RCC controlled oscillators can maintain global multi-stable states, and domains where power-law relations may emerge. The network of oscillators deterministically stabilises into different orbits for different perturbations, and the relation between the perturbation and amplitude can show exponential and power-law correlations. This can be considered to be representative of a basic mechanism of protein production and control, that underlies complex processes such as homeostasis. Providing feedback from the global state, the total system dynamic behaviour can be boosted or reduced. Controlled SOC can provide much greater understanding of biological control mechanisms, that are based on distributed local producers, with remote consumers of biological resources, and globally defined control.

## Introduction

Our physical universe is inherently nonlinear, chaotic and noisy. This makes it difficult for biological systems to exist because they require stability to function. The concepts of dynamic stability in complex biosystems is generally accepted [1], but not very well understood at all organisational levels. The capacity of biosystems to control their internal state and their

**Competing interests:** The authors have declared that no competing interests exist.

external response to the chaotic environment may be contributed to their ability to adapt in a nonlinear manner. The facility to quickly change their dynamic state, apparently scale-free, is essential to the survival of each organism [1]. Furthermore, the high level dynamic stability of a larger organism must be compiled from the lower-level dynamic behaviour of many contributing elements. In particular, the local information available to each organ, cell, sub-cellular body, or other organic organisation, is severely limited with regards to the global whole body state, and is also limited in its ability to respond in a timely and measured manner. Here, the global state is meant to be interpret as the total state of all the contributing elements, such as molecules, cells or even organisms. Although there exists some robust understanding of the mechanisms involved in global stability of an organism [2], this is certainly not the case for the required complexity of local small elements collaborating in a nonlinear biological system to generate a globally stable dynamic state [3].

## Criticality in biosystems

To allow a biosystem to control its environment using only local control at its disposal, the system must be able to exert a nonlinear effect on its global behaviour. To achieve this multiplicative effect of each local contribution, the behaviour must be able to show exponential, or even power law relation between its local behaviour and the global response [4]. This may well be achieved using Self-Organised Criticality (SOC), a concept that allows a nonlinear system to show emergent power law relations between the local behaviour and the global state. There exists a suitably long history of identifying possible power law relations in biology, and debates whether or not it exists, or is even functional [5]. Subsequent efforts have been made to create such an emerging SOC system based on biological facts, from early attempts to explain this concept in insect populations [6], epidemiology [7], neuronal behaviour [8, 9], or engineered concepts that are based on relevant data [3, 10, 11]. It is essential to separate the various elements related to SOC, which is not an intuitive concept to grasp, and can be regarded from different angles depending on the view of interest for the researcher. One may find in literature a wide range of topics that reflect these points of interest [12], the means of determining its existence within a data set [5, 13], its relevance to biophysical systems [14], and other possible controversial aspects of its existence and meaning [15].

## Self-organised criticality

Self-Organised Criticality is a distinct property of physical systems based on the local interactions of many small components, each of which contributes to the global critical system [16, 17]. Depending on the field of study, there seems to exist different interpretations of the concept of Self-Organised Criticality [18], but the concept of *critical behaviour*, or just *criticality*, have been neatly summarised by Watkins et al. [15]. Irrespective of alternative ways of generating SOC systems, the proposed mechanism within this paper is not based on a physical phenomenon as such, and focuses on the mathematical interpretation of SOC systems within dynamic systems theory. This is described by the ability of complex systems to reside at an *equilibrium point* or *critical point* within their parameter space, such that they will change from one state to another with only a small perturbation to the system [19]. Therefore, weak perturbations of the system from external sources may cause a state of change due to the critical point, where the system will evolve into the new stable or unstable state. The proposed mechanism of control ensures that the system remains stable, in the Lyapunov sense [20], where the local neighbourhood defines the stable domain to which the system will return when perturbed.

Small interactions between the elements of a SOC system are not necessarily large with respect to the contribution of each of those elements, but they add to the global critical state

due to nonlinear behaviour. These nonlinear local perturbations are usually observed as rapid transitions from one state to another. A typical illustration is an avalanche of snow or sand, where a previously apparent stable state rapidly changes to a new stable state. Similar critical dynamics have been determined in biological systems, at low [21] and high levels [22], and in particular in neural dynamics [23, 24]. The functional role of criticality, the self-organised and self-sustaining multi-stable state of the system, appears to be mostly related to network complexity, and power-law relations within those networks [25]. However, it is still unclear if power-law relations are a true property of large scale complex systems over the entire scalar domain [26]. Furthermore, as will be shown, the emergent power-law relations may be considered only an epiphenomenon of the combined response of the nonlinear oscillators, and may exist for only part of the parameter domain. In this domain, the essential property of the system is stable periodic behaviour throughout these perturbations. For the described systems below, the means by which the criticality may emerge is not based on the separation of time-scales, but on the local adjustments by the RCC control mechanism in combination with nonlinear connectivity. The proposed method contains all three key features for a system to be in a critical state, namely non-trivial scaling due to the external perturbation, spatiotemporal power-law correlation in respect to the total behaviour of the network, and self-tuning to the critical point where the network self-selects the periodic orbit [15].

## Properties of SOC within biosystems

Considerable effort has been put into deciding the characteristic properties of criticality in large and complex systems. Aspects, such as connectivity, and dynamics are known to be important, on which the principle of Artificial Neural Networks is based. Here, I argue that the problem can be said to lie in determining how a local system, usually at a much smaller scale, can contribute to a global, stable system of which they form only a small part. Within dynamic systems theory, these local systems can act as perturbations of the overall global system. Even if the individual elements are themselves dynamically stable (in a steady state), the entire system may become unstable due to these perturbing elements. This is commonly seen in spatiotemporal chaos, where individual elements are in steady state, yet they destabilise the entire system when weakly coupled [27]. Within biosystems this property, where clusters of smaller elements contribute to the total behaviour, is very common. For example, it may be seen clearly in the organs of large multi-cellular organisms [28, 29]. Loss of stability in any of these localised systems may be catastrophic for the entire global system. The ability to ensure that the entire system remains stable may not depend on a specific supervisory element, but must be provided by the cooperative nature of the constituting elements. This, in effect, precludes the use of supervisory control, even if that is a much higher order controller, such as the brain [30]. This characteristic property of biological systems to exhibit emergent dynamic behaviour is based on local small scale distributed dynamics behaviour which collaborates to produce a higher order dynamics. In particular, the ability to respond in a characteristic nonlinear manner, such as a log scale response to input [31], as well as the regulation of producing and consuming biomatter. Different parts of the organism are involved in maintaining and developing the global system and these small conglomerates of cells or organisms can produce types of activity similar to the behaviour of larger organisations [1]. For these disparate systems to cooperate effectively, it would seem that local control is not sufficient. However, if these systems share properties with critical systems [14], it becomes feasible that by changing the response of the local system to the external conditions, a suitable state can be found that is appropriate for control to be effective at the global scale. This means that the proposed mechanism of control permits local control of homeostatic systems without external or supervisory approaches.

## SOC and homeostasis

Homeostatic control is based on negative, as well as occasionally positive, feedback of some of the components in the feedback loop in relation to some set point. This well known concept is enticingly simple, and has echoes of the older concepts of physical harmony, and well balanced systems. It exists in many biological systems at almost all levels, from sub-cellular molecular dynamics [32, 33], through physiological phenomena (e.g. glucose and blood pressure levels) [34], neural dynamics [35], and all the way to sociological behaviour. Both from experimental data and from physiological experiences [36], this concept seems to be somewhat flawed. In dynamic systems theory it has been shown that feedback systems can only under the strictest circumstances be guaranteed to be stable. For example, it is shown that a reduced system can have multiple stable states using isolated feedback loops [37] but only for a specific parameter set. Positive and negative feedback loops may have both stabilising and destabilising effects, depending on the shape of the Jacobian matrix for only some determined stable states [38]. Lastly, even an unified control approach, emphasising the possibility of hierarchical control, avoids the issue of system stability and does not consider the effect of the complexity of each of the boxes connected within various feedback loops [2].

It has become clear that even well known homeostatic systems are very rarely stable in the dynamic sense, appearing to have multiple stable states, and do not show the properties of stable systems when perturbed. This has now been encapsulated within the concept of allostasis that permits multiple stable states within an homeostatic system [39]. Assuming that a homeostatic system is the apparent result of underlying stabilising mechanisms, rather than the mechanism itself, it may be possible to explain this concept, and its limitations in a consistent dynamic systems manner, using criticality [14, 21]. Employing criticality to understand dynamic interactions allows the system to be highly variable, as indicated by multiple stable but controlled states, and at the same time be dynamically stable in the sense that each individual state does not destabilise the system. Providing dynamic stability with high variability of the emerging system will allow for a better understanding of the underlying mechanisms, and may provide possible ways to resolve the limitations of homeostatic control that currently cannot be addressed appropriately [40].

Within the Methods section, it is explained how SOC is constructed based on networks of RCC controlled oscillators. The Results section discusses weakly connected pools of these oscillators to demonstrate that the use of localised control by RCC allows the entire system to be globally stable. It also shows the, often considered essential property of SOC, of scale-free behaviour that emerges from the overall dynamics. Finally, it is shown how this approach allows SOC to underlie the basic concepts of homeostasis within a biological organisation, providing a means to understand these complex relations as a biodynamic behaviour.

## Methods

Recently, it has been shown that nonlinear chaotic dynamic systems can be stabilised using the Rate Control of Chaos method. This method adjusts the rate of evolution of part of a nonlinear system such that the exponential growth of an unstable chaotic oscillator is controlled into stable limit cycles. The control is determined using the rate of growth of some of the variables in proportion to the overall embedded phase space of those variables. This is then applied to an exponential control function that, in effect, causes the rate of change of the variable to be controlled to speed up, or slow down. The proportional rate of change is unity when no control is applied or when the system is not changing exponentially [41, 42].

This method may be regarded as an extension of the traditional biochemical enzyme control concept by adjusting the reaction rate. The ability to control the stability of biochemical

reactions, by controlling the rate of reaction based only on local information, allows the biosystem to function under a wide range of conditions. Furthermore, it has been shown that this mechanism can be extended further to control higher level spatiotemporal chaos, even if the underlying dynamics of each element is chaotic [42]. The individual elements, as well as the total observable system, are stable in the sense of Lyapunov stability. This means that the system will reliably return to the same area of phase space relative to its controlled dynamics, although it does not necessarily follow that this is the same as the uncontrolled chaotic domain. Therefore, the RCC method does not eliminate entirely the chaotic properties of the underlying nonlinear system, but applies limited localised control to the system to maintain an apparently stable system. The controlled system still has many properties of the nonlinear system; it can respond nonlinearly to perturbations and can be weakly chaotic. This is illustrated in Fig 1A, where the RCC controlled system of a bienzymatic model, described below, is controlled into a stable orbit. In Fig 1B is shown the local Lyapunov estimates for the controlled system, demonstrating weak chaos. The phase space of both the controlled and uncontrolled system can be seen in Fig 1C.

The nonlinear model of a bienzymatic cycle used in this paper by Berry [43], described by (4) to (7), has been shown to be controllable using the Rate Control of Chaos (RCC) method, such that the control allows the stabilisation of the external environment by adjusting the amount of enzyme based on the local amount of one of the components $f$ [44]. This model describes the two enzymes that control the formation of extracellular matrix $m$ from soluble filaments $f$. The proteinase $p$ transforms the matrix into filaments, and the transglutaminase $g$ converts the filaments into matrix. Extracellular matrix is produced by neighbouring cells $r_{im}$ at a constant rate, and each protein decays in catalytic processes proportional to $p$. $r_{im}$ is a bifurcation parameter, that may cause the system to become unstable and chaotic. Application of the RCC method, described by the quotient $q_f$ (1), and the two control functions $\sigma_p$ (2), and $\sigma_g$ (3), controls the production of the two enzymes $p$ and $g$ (in (6) and (7)) such that the system remains in controlled stable orbits for large ranges of values of the bifurcation parameter $r_{im}$. Within the subsequent simulations, the model is shown as time series of the main variables $m$ and $f$ or phase space plots of $f$ (x-axis) versus $m$ (y-axis).

$$q_f \quad = \frac{f}{f + \mu_f} \tag{1}$$

$$\sigma_p(q_f) \quad = f_p \, e^{(\xi_p \, q_f)} \tag{2}$$

$$\sigma_g(q_f) \quad = f_g \, e^{(\xi_g \, q_f)} \tag{3}$$

$$\frac{dm}{dt} \quad = k_g \frac{f\,g}{K_G + f} - \frac{m\,p}{1 + m} + r_{im} \tag{4}$$

$$\frac{df}{dt} \quad = -k_g \frac{f\,g}{K_G + f} + \frac{m\,p}{1 + m} - \frac{f\,p}{1 + f} \tag{5}$$

$$\frac{dp}{dt} \quad = \sigma_p(q_f)\gamma \frac{f^n}{K_R^n + f^n} - k_a p^2 \tag{6}$$

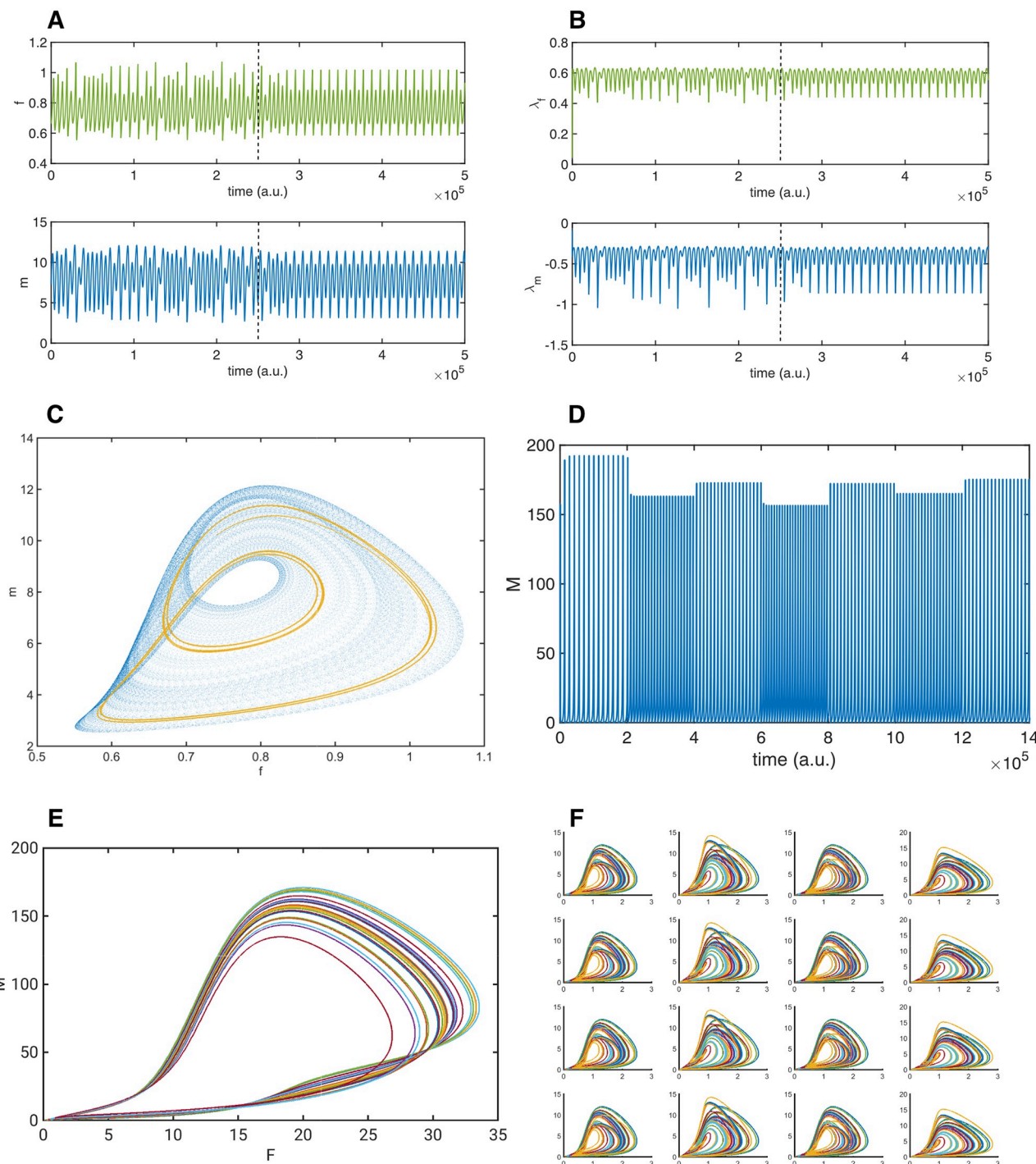

**Fig 1. Dynamic controlled behaviour of the Berry model.** *A* Example of the chaotic Berry model with control enabled at the dotted line, the system quickly stabilises into a two orbit. Top panel shows the modelled soluble filament *f*, the bottom panel the fixed matrix *m* in time. *B* Lyapunov estimates of the model in Fig 1A, showing small positive and small negative values. The control, enabled at the dotted line, does not eliminate the chaotic state, but changes the system into a stable oscillation. *C* Phase space plot of the chaotic model in blue, with the RCC controlled two orbit superimposed in gold, *F* versus *M*. *D* Total unweighted sum *M* from sixteen weakly coupled RCC controlled oscillators, with random external perturbations at fixed intervals showing 7 stable oscillations after very short transients. *E* Phase space plot *F* versus *M*, of 24 stable oscillations based on the total unweighted sum of the sixteen weakly coupled oscillators. Each coloured line is a single stable oscillations, as can be seen in time in Fig 1D. *F* Phase space plots of the sixteen systems *F* versus *M*, where each coloured line is a single stable oscillation, different due to the random external perturbations (transients removed).

$$\frac{dg}{dt} = \sigma_g(q_f)\beta \frac{f^l}{K_S^l + f^l} - k_{deg} \frac{gp}{K_{deg} + g} \tag{7}$$

The Berry model parameters are as follows; $\gamma = 0.026$, $\beta = 0.00075$, $K_R = 4.5$, $K_S = 1$, $K_G = 0.1$, $K_{deg} = 1.1$, $k_g = k_{deg} = 0.05$, $k_a = \frac{k_{deg}}{K_{deg}} = 0.0455$ and the Hill-numbers $l = n = 4$. For different values of the bifurcation parameter $r_{im}$ in (4), the model exhibits a wide range of dynamic behaviour, including periodic cycles, bistability and chaos [43]. This parameter is kept for all oscillators within the chaotic domain. External input is provided to this parameter in the perturbation experiments (8), and this parameter is used to connect the different oscillators together using a scaled relative contribution from all the other oscillators (i.e. no self-connections):

$$r_{im}^i = \sum_{k=1, k \neq i}^{n} w_k m_k + \epsilon \tag{8}$$

where $w_k$ the connectivity strength for the oscillator which can be either 0.00011, 0.00012, or 0.00025, and remains within the chaotic domain. $\epsilon$ is a uniform distributed perturbation term drawn from the domain $[-1, 1]$ and scaled to the connectivity strength of each unit. The perturbation used for the oscillators is the same random uniform distributed value applied to each oscillator, but is scaled by the randomly chosen values [7.5, 1, 8, 3.25], although each column in Fig 1F has the same perturbation value to ensure that the network is not symmetric. Furthermore, the perturbation is redrawn after a certain number of evolution steps, allowing the system to be explored for different values of perturbation, resulting in different oscillations. The connectivity strength values for each oscillator may vary, and this affects the dynamics, but not the stability. This is shown in the simulations with stepwise increase of the connectivity strength, starting from $w_k$ = [0.0001, 0.0002, 0.0003, 0.0004] for each column, and increased after $10^5$ time steps by 0.00005 (Fig 2C). The connectivities were therefore kept within the chaotic domain of the underlying oscillators, and further work is under way to demonstrate the utility of the connectivity strength within these models. The RCC method can be shown to stabilise spatiotemporal patterns, which may become unstable due to local nonlinear interaction, and is effective even when the underlying systems are chaotic [42]. The advantage of this method is that it allows nonlinear systems to be stabilised into periodic stable dynamics based only on the local dynamics of each individual oscillator. The RCC parameters were also kept constant in these models, but can be varied to change the shape of the local oscillator. For the first perturbation experiments (described in Figs 1–4), the control parameters in (1) are $\mu_f = 2$, and in the RCC functions for each of the enzymes $p$ (2) and $g$ (3) are $\xi_p = -1$, $f_p = 1$, $\xi_g = -1$, $f_g = 1$. In the criticality experiments with global feedback, the control is enhanced by $\xi_p = -3$, and $\xi_g = -3$, with no other changes. In the experiments with 64 oscillators (see Fig 4), the connectivity strength for each oscillator was a randomly assigned unique value between 1 and 10. This was to show that the parameters do not need careful tuning beyond ensuring that the system is in the chaotic domain, and RCC controlled.

Using simple finite-difference connections between the oscillators, or more elaborate connectivity schemes as required, the critical dynamics can be generated. The individual oscillators will adjust their local dynamics to accommodate the perturbations by their immediate neighbours. The global dynamics may then become critical due to the nonlinear oscillators. As shown in the results, imposing a linearisation method on the connectivity scheme, such as Crank-Nicholson discretisation, will remove the high frequency nonlinear behaviour and results in the loss of criticality of the global state. The network of oscillating nonlinear models

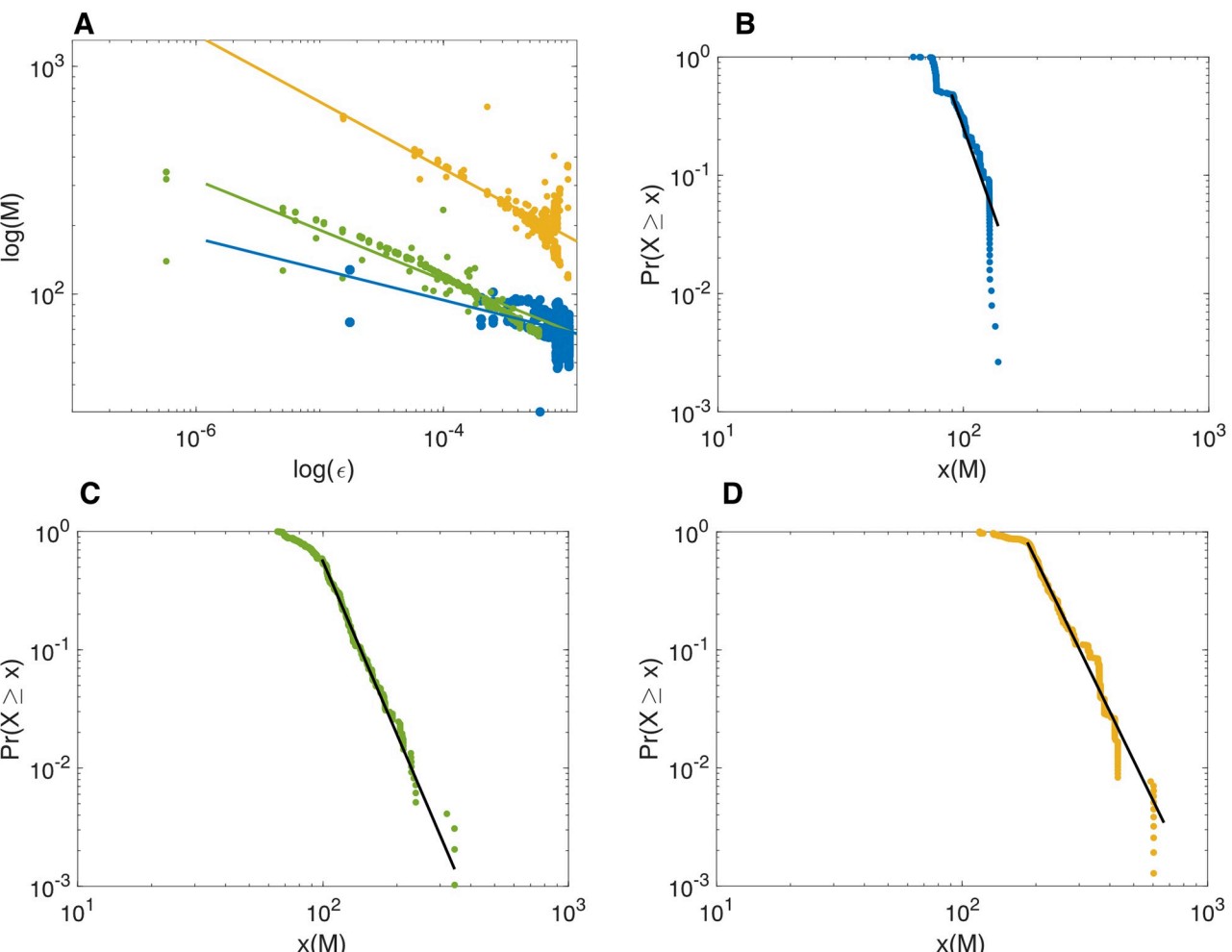

**Fig 2. Network size in relation to power law due to random perturbations.** *A* Log-log plot of the total unweighted sum *M* from 8 (blue), 16 (green), and 32 (gold), weakly coupled RCC controlled systems, with random external perturbations. The power curve fits are $M(8) = 26.88\epsilon^{-0.14}$, $M(16) = 15.19\epsilon^{-0.22}$, and $M(32) = 32.48\epsilon^{-0.29}$. *B* Power-law estimation function for the 8 coupled oscillators with a minimum of $M = 64.5$, and slope $\alpha = 7.05$. *C* Power-law estimation function for the 16 coupled oscillators with a minimum of $M = 99$, and slope $\alpha = 5.85$. *D* Power-law estimation function for the 32 coupled oscillators with a minimum of $M = 184.5$, and slope $\alpha = 5.27$.

is simulated using standard ODE fixed step integration methods. The fourth-order Runge-Kutta (RK) method, as well as the higher order extensions, Fehlberg RK and Prince-Dormand RK, provide suitable results with varying choice of integration step size. The qualitative results are readily reproducible using any of these methods. The simulation software EuNeurone, used for these models, is available as open source, as well as the model equations. Both are available on Zenodo [45, 46].

The total unweighted dynamics is determined by the sum of the individual oscillators *i* for a total of *n*, as could be seen by a remote observer for whom the individual oscillators are not readily visible.

$$M \;\; = \sum_{i}^{n} m_i \qquad (9)$$

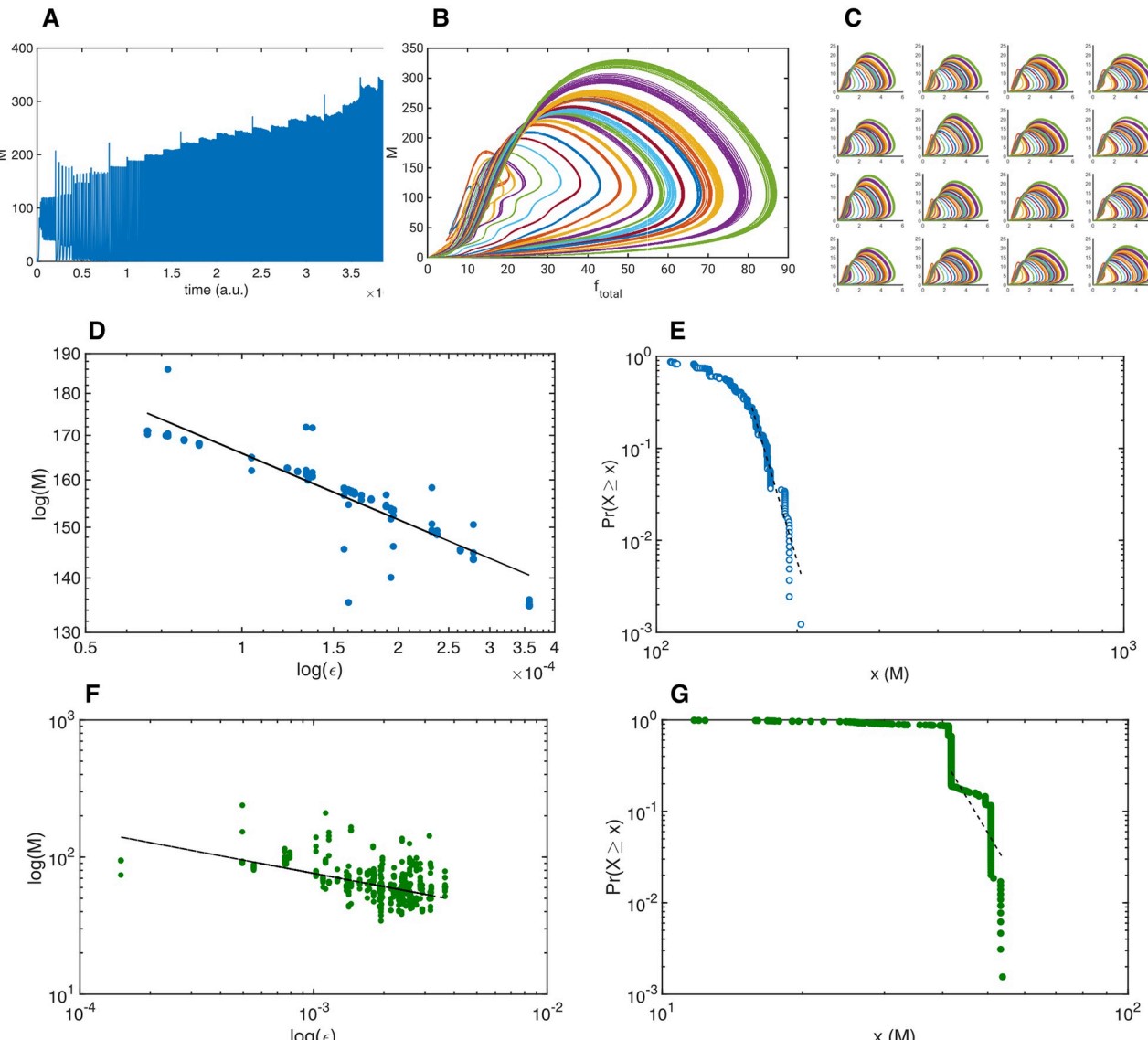

**Fig 3. Importance of connectivity in creating an RCC critical system. *A*** Total unweighted sum $M$ from sixteen weakly coupled RCC controlled systems, with random external perturbations, as well as, incremental increases in the connectivity strengths. ***B*** Phase space plot of the total dynamic behaviour of the perturbed systems, with incremental stepwise increases of the connectivity strengths. Each coloured line is one stable oscillation. ***C*** Phase space plots of the sixteen weakly coupled individual systems, with incremental increases in connectivity strengths. ***D*** Log-log plot of the external perturbation $\epsilon$ versus the maxima of the total unweighted sum of $m$. Fitted power curve in black ($3.9\epsilon^{-0.13}$). ***E*** Power-law estimation of the maxima of the network with the fitted power function (dotted line) superimposed with minimum $x(M) = 160$ with slope $\alpha = -18.24$. ***F*** Log-log plot of the external perturbation $\epsilon$ versus the maxima of the total unweighted sum $m$ with Crank-Nicholson interpolation connectivity scheme. The fitted power curve in black $2.1\epsilon^{-0.32}$. ***G*** Power-law estimation of the maxima of the Crank-Nicholson connected network with the fitted power function (dotted line) superimposed; minimum $x(M) = 41.76$, slope $\alpha = -9.55$.

$$F \quad = \sum_{i}^{n} f_i \tag{10}$$

The summed total of the variables $m$ for each of the oscillators is added to Eq (8) for the models with global feedback, scaled either negatively or positively as appropriate ($\nu_k =$

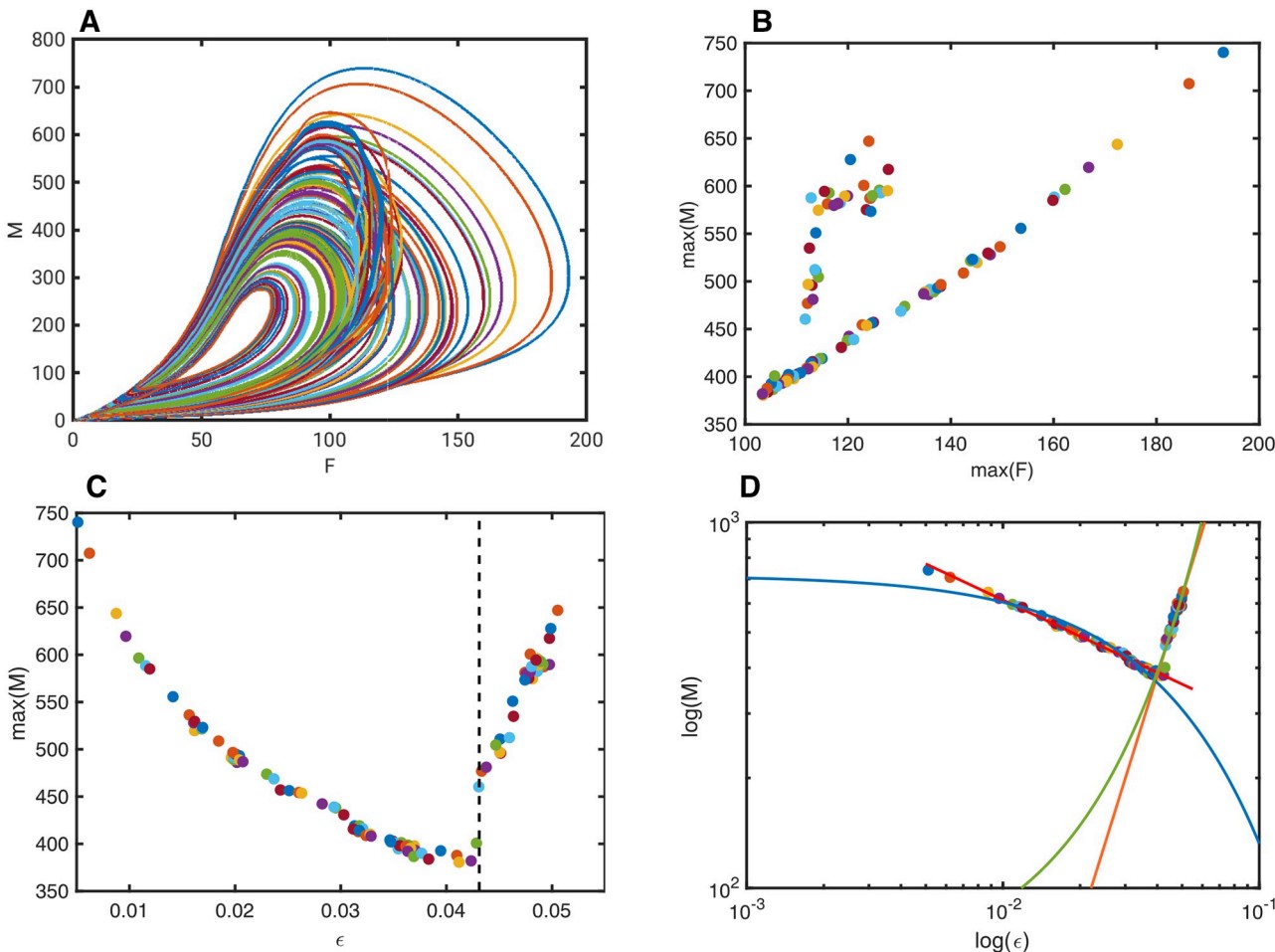

**Fig 4. Domains of power law relations within perturbation space of RCC controlled Self-Organised Criticality. *A*** Phase space plot of *M* versus *F* of 100 stable oscillations in a 64 oscillator network. ***B*** Plot of the maxima of *F* versus the maxima of *M* of the 100 orbits on the left. Notice the two domains where the dynamics of the oscillators change. ***C*** Plot of the Normalised Gaussian distributed perturbation $\epsilon$ with variance 10 versus the maxima of *M*, showing that around 0.043 the oscillators change their shape. ***D*** Log-log plot of the Normalised Gaussian distributed perturbation $\epsilon$ versus the maxima *M*, same as the panel on the left. Additionally, the curve fits for the power functions $134.7\epsilon^{-0.328}$ (red), $5.38 \times 10^5\ \epsilon^{2.254}$ (orange), and exponential function fits $715.9e^{-16.89\epsilon}$ (blue), $56.59e^{48.26\epsilon}$ (green). This shows clearly that the system can have a power law relation for limited size perturbations, and a non-power law relation otherwise.

±0.00001), to ensure that the individual oscillator is within the controlled domain. This feedback is added during the evolution of the differential equations, so that *M* is due to the instantaneous value determined at each integration step, and therefore not merely equal to the current summed input of the previous time step reflecting the local instantaneous state of the system.

$$r_{im}^i = \sum_{k=1, k\neq i}^{n} w_k\, m_k + \epsilon \pm v_k M \quad \text{where} \quad v_k < w_k \tag{11}$$

The Eq (11) shows that feedback for this model in effect reduces the connectivity as provided to $r_{im}$ for each oscillator, but because of the weaker feedback, it does not disrupt the network such as much stronger connectivity or little connectivity would effect.

The log-log plots are generated by the maximal peak values of the unweighted sum max(*M*) during a single oscillation, and used to determine the power plot as estimated using the power-

law estimate function according to Clauset et al. [13]. This method estimates the scaling parameter $\alpha$ from the power law probability distribution $p(x) = Pr(X = x) = Cx^{\alpha}$ using the method of maximum likelihood. This is determined using the numerical maximisation of the logarithmic likelihood function described by

$$\mathcal{L}(\alpha) = -n \ln \varsigma(\alpha, x_{min}) - \alpha \sum_{i=1}^{n} x_i \tag{12}$$

The minimum bound for $x_{min}$ for these estimates is determined by the Kolmogorov-Smirnov statistic, that is the maximal distance between cumulative distribution functions of the data and the fitted model. It is noted that these methods would require a significantly large data set for reliable estimation. Within the shown simulations, these estimates are provided primarily to demonstrate that the overall probability of a power law relation is feasible.

Determining the fitness of any model that can describe the datasets is more readily achievable for probabilistic approaches that allow the preplanned comparisons between models. The goodness of fit, as used above, employing the method of maximum likelihood, is only appropriate to the extent that it allows the model to be reconciled with the options available, in this case the existence of some power law relation. It also suffers somewhat from the ability to be a suitable predictor due to possible overfitting. Similar complexity measures for model estimates are the Bayesian Information Criterion (BIC) and the Akaike Information Criterion (AIC) that can indicate the best fit of log-likelihood, often used for predictors in regression models. These can be determined as follows, based on the Mean Squared Error.

$$aic = -n \ln \left( \frac{1}{n} \sum_{n=1}^{n} (x_i - \hat{x}_i)^2 \right) + 2k \tag{13}$$

$$bic = -n \ln \left( \frac{1}{n} \sum_{n=1}^{n} (x_i - \hat{x}_i)^2 \right) + k \ln(n) \tag{14}$$

where $k$ is the number of parameters of the estimated model, and $\hat{x}_i$ is the estimated fit. This is used below to demonstrate the deterministic nature of the method of SOC generation and the suitability of the model estimates.

The generic properties of a two enzymatic control of a producer and consumer system, such as the formation of (extra-)cellular matrix of the Berry model, can be considered to be representative of many biological processes where a resource is controlled by antagonistic control using two enzymes. The model is not considered to be solely representative of this concept, but is recognisably typical for many types of control needed for biological control processes. The Rate Control of Chaos has successfully been used to control many different models, e.g. the Rössler and Lorenz models [41], but also the Grey-Scott [27], and the Lengyel–Epstein models [47], that have been used to model biologically relevant patterns of insulin release in a criticality based system.

## Results

### Local control provides global stability

To illustrate the emerging global controlled critical behaviour, a simulation of 16 coupled RCC controlled chaotic oscillators is used. These oscillators receive local random external perturbations (i.e. external to these oscillators), causing them to change their local dynamics in response. To visualise this, in Fig 1D is shown a short series of only 7 perturbations, where the

perturbations are randomly varied every 20,000 time steps. Note that after only very short transients, the total global behaviour of the unweighted sum of the oscillators becomes stable in a different oscillation each time. The phase space plot of a sample of 24 randomly perturbed oscillations is shown in Fig 1E, where the total unweighted sum of the two main variables of each oscillator are plotted, with the transients removed. Each different coloured line represents one of these stable oscillations, showing changes in the dynamics, but only depending on these perturbations. The individual oscillators that combine to generate the global stable dynamics are shown in phase plots in Fig 1F, where each coloured line represents the stable oscillation under the perturbed conditions, with transients removed. As explained in the Methods section, Fig 1A, shows the RCC controlled system of a bienzymatic model controlled into a stable orbit. In Fig 1B is shown the local Lyapunov estimates for the controlled system, demonstrating weak chaos. The phase space of both the controlled and uncontrolled system can be seen in Fig 1C.

The number of oscillators in the network naturally affects the amplitude of the summed oscillators, but the critical multi-stability property of the network is preserved. This is shown in Fig 2A, where the log-log plot of three networks are shown. The lower graph shows total behaviour $M$ of 8 coupled oscillators, the middle of 16 oscillators (same network as shown in Fig 1, but with different random perturbations) and on top, 32 coupled oscillators. The curve fits that are superimposed on the three sample behaviours show the power relations $M(8) = 26.88\epsilon^{-0.14}$, $M(16) = 15.19\epsilon^{-0.22}$, and $M(32) = 32.48\epsilon^{-0.29}$. To show that the underlying relation has some power law relation rather than an exponential or similar relation, the power law estimation functions for each of the three samples of coupled networks were determined (see Methods). In Fig 2B for the 8 coupled oscillators with a minimum of $M = 64.5$, and slope $\alpha = 7.05$. In Fig 2C for the 16 coupled oscillators with a minimum of $M = 99$, and slope $\alpha = 5.85$, and in Fig 2D with a minimum of $M = 184.5$, and slope $\alpha = 5.27$. Due to the limited number of oscillators used, the power law function does not cover many decades. The results show that the near power law relation holds for several sizes of networks; large scale modelling, which is computationally expensive, will need to show the full range of scale-free behaviour.

By modelling a simple network of controlled chaotic nonlinear oscillators, it can be seen that even with random perturbations, the control allows the total system to remain stable. It adapts to these perturbations by stabilising into different orbits. The perturbations can still destabilise the entire system if their contribution is disproportionally large with respect to the ergodic properties of the individual oscillators. The number of elements affects the dynamics by allowing higher amplitudes, but more importantly the system maintains multi-stable states and has apparent power law relations, even for such small networks.

## Scale-free emergent behaviour

To demonstrate how the network of rate controlled oscillators is capable of generating apparent scale-free behaviour, the same network is used, still randomly perturbed by uniform perturbations on each of the oscillators, but simultaneously the network connectivity is stepwise increased. Starting with a relatively low connectivity, strong enough to maintain some cohesion between the oscillators, the connectivity between the oscillators is increased stepwise at a constant rate. This is shown in Fig 3A, where the unweighted sum $M$ is shown in time, with stepwise increments of connectivity at each multiple of $10^5$ time steps. Even though the dynamics of the total system may change, it remains stable throughout. Not shown is that if the connectivity becomes too large, $M$ will explode to a singularity. In Fig 3B is shown the total dynamics of $f$ versus $m$ of each individual stable oscillation indicated by individual different colours, with transients removed. Although at higher connectivity strengths the orbits are

more noisy, due to the amplification of the individual random perturbations, the total system remains stable. This can be seen in Fig 3C, where the sixteen oscillators are shown, for each of the incremental steps, clearly demonstrating their individual stability, even at high connectivity.

The changes in connectivity could for most coupled dynamic systems be a source of instability, often causing bifurcation phenomena. However, the local control manages to stabilise each individual oscillator, based on local information alone, and thereby ensures that the global system remains stable as well. The subsequent total or global system also demonstrates properties of criticality, in that the total system amplitude grows much faster than merely the sum of the individual elements a linear (or reduced) system would show. To quantify this, the peaks $M$ are determined (the maximal value of $M$ once stabilised past the transient), and plotted versus the amount of local perturbation in Fig 3D. To demonstrate independence of the connectivity strengths of this scale-free property, the data from the perturbation simulations are used, as in Fig 1D. The log-log plot shows the apparent power-law relation, with a fitted power curve $3.9\epsilon^{-0.13}$. The corresponding power-law estimation function is in Fig 3E, with a minimum of $M = 160$, and slope $\alpha = 18.24$, demonstrating that the data is partly power distributed. It is interesting to observe that this scale-free aspect results in an almost power-law relation which is characteristic for biological observations [26].

Testing the hypothesis that the emergent critical system is based on the combination of the RCC controlled chaotic system with weak to moderately strong connectivity, the connectivity function was modified to match the standard Crank-Nicholson interpolation method, using a two-by-two stencil of the local neighbourhood. The peaks of $M$ were determined based on the uniform perturbed network of oscillators, and plotted against the perturbation in Fig 3F. No system parameters were modified, apart from the interpolation connectivity; therefore both the individual oscillators and the total system remained stable, but have lost the scale-free power relation. The power-law estimation function for those data samples is in Fig 3G, with a minimum of $M = 41.76$, and slope $\alpha = 9.55$, showing no power law relation present.

To further investigate the effect of perturbations on a larger pool of oscillators, a network of 64 oscillators was individually perturbed with Normalised Gaussian Distributed perturbations with mean of 0.00005 and variance of 10, to ensure that each oscillator is receiving sufficiently different input. The choice of Gaussian distribution to replace the uniform distribution is made to show that the approach is independent of specific distributions for criticality to emerge. The resulting stable oscillations of 100 orbits are shown in a phase space representation in Fig 4A, where each colour represents a different stable oscillation of the total behaviour of the summed variables $F$ versus $M$. In Fig 4B can be seen the maxima of each stable oscillation of $F$ versus $M$, showing that there are two distinct domains of oscillatory behaviour emerging from this network. The first type is a single orbit that seems almost linearly related. The second type is a two-orbit that causes an apparent cluster on the left, above the apparent line. However, the underlying relation between the perturbations and the effect on the total dynamic behaviour is not linear, as can be seen in Fig 4C, where the total summed perturbations of the 64 oscillators is plotted against the maxima of $M$. It becomes clear that the left part of this plot for $\epsilon <= 0.043$ (before the dotted line), exhibits a power-law relation between the total perturbation $\epsilon$ and the maxima of $M$. This is shown in a log-log plot in Fig 4D, where superimposed are also shown four curve fits. The power function $134.7\epsilon^{-0.328}$ in red is fitted to the curve for $\epsilon <= 0.043$ and clearly expresses the sub-domain where a power law relation exits. In blue is shown the fit of an exponential function $715.9e^{-16.89\epsilon}$ on the same domain, which is clearly quite poor. The domain for $\epsilon > 0.043$ fits a power law $5.38e + 05\epsilon^{2.254}$ in orange) but also an exponential function in green $56.59e^{48.26\epsilon}$, indicating that this domain does not have a power law relation, and may be exponential or simply linear.

These models are based on deterministic behaviour of the perturbed network which itself is based on controlled chaotic oscillators. The models are only perturbed from one state to another by the random perturbations, and no noise is included in these systems. The resulting behaviour is therefore fully stable deterministic and is completely described by the RCC control. It is possible to determine the log-likelihood of the data for model selection. The relevance of probabilistic model selection approaches is very limited to this type of modelling because statistics such as AIC are designed for preplanned model estimates that do not take into account the model parameter space, which for a large set of nonlinear differential equations is extensive [48]. It should also be considered that the underlying model is not based on probabilistic methods, and that the emerging power or exponential relation causes great variance in the data, which greatly amplifies the mean squared error. As a representative example of such estimates, the BIC and AIC were determined for both the power fit model, and the exponential fit model on the data shown in Fig 4D.

The model fits with samples for $\epsilon < 0.043$ are $135.5\epsilon^{-0.3277}$, and $712.7e^{-16.65\epsilon}$. In this case, for the power fit, the Mean Square Error is 46.2, BIC is −247.98., AIC is −254.64; and for the exponential fit MSE is 374.74, BIC is −390.33, AIC is −396.98. Despite the relatively large value of MSE, the power fit still seems better suited than the exponential fit. For the model fits with samples $\epsilon > 0.043$, the fits are $5.387e+5\epsilon^{2.254}$, and $56.59e^{48.26\epsilon}$. The corresponding values for the power fit, MSE is 304.47, BIC is −183.85, AIC is −188.43; and for the exponential fit, MSE is 330.63, BIC is −186.65, AIC is −191.23, which shows there is little difference in these estimates for model selection, possibly favouring the power fit somewhat.

It can also be concluded from this set of simulations that the relation between perturbations, the number of oscillators, and connectivity strengths is not straightforward. However, the resulting system of connected oscillators is dynamically stable, can exhibit different types of emerging relations between the total amplitude of the oscillators and the perturbations, and also that the emerging power law relations are an epiphenomenon of the network's attempt to stabilise its overall behaviour in response to external random perturbations.

The described network of nonlinear RCC controlled oscillators is also quite noise robust. The perturbations cause the complete system to assume a specific orbit, if these perturbations themselves also vary over time, this will result in continuously adjusting orbits, but within the same deterministic neighbourhood. This property is derived from the original RCC control that aims to maintain the orbit in a stable oscillation (see [42]), and can be understood by considering that the control will correct for noise in the same way as it does for chaos.

The apparent power-law relations found within a domain of the perturbation parameter provide a strong indication that the nonlinear behaviour of each oscillator causes perturbations such that the total behaviour of these oscillators is much stronger than the individual contributions they provide. This is not based on the onset of synchronisation due to changing coupling strengths, because it occurs for constant coupling and there is no critical coupling strength for which the relation holds.

It has already been shown that chaotic units may generate a critical system [49], where the effect of connectivity over time is shown to have a power-law relation between connectivity and the connected chaotic systems. Here, the connectivity level of $k$-connectedness is critical to establish the power law stability. Therefore, a locally controlled mechanism of generating critical rate controlled systems, as described in this paper, may provide the necessary dynamics for complex interactions found in biological systems. For example, local producers of a protein can regulate their behaviour on local information alone, but still provide the effect needed by remote consumers of the protein. This may be particularly important for regulatory processes and may provide the key elements of a feedback loop process that responds rapidly to changing global behaviours.

## Criticality as a homeostatic process

Adapting the network of sixteen coupled oscillators to include a global feedback of the total dynamic behaviour is possible by adding the scaled total behaviour to the external perturbation of each oscillator, ensuring that the perturbed input to each of the oscillators is not too great to push it out of the controlled domain. By making this feedback either positive or negative, the effect of global input to the individual oscillators as negative and positive feedback loops is shown. In Fig 5A is shown the total dynamics of the network, with stepwise increased connectivity and positive feedback, but without further random perturbations. In this case, the different stable orbits due to the increased connectivity are amplified to higher totals than without the positive feedback. The corresponding power-law estimation function, based on the maximal values of these orbits, shows a strong power relation between the connectivity and the total amplitude of the oscillations (Fig 5B). Conversely, if the feedback of the total behaviour is negative, the total dynamics is more compressed, as can be expected (Fig 5C). Also, the individual oscillations are more limited, with less drift. The power-law estimation function of the negative feedback (Fig 5D) shows, interestingly, that the critical system has lost its power-law relation and appears more linear. This would suggest that, just like in classical homeostatic

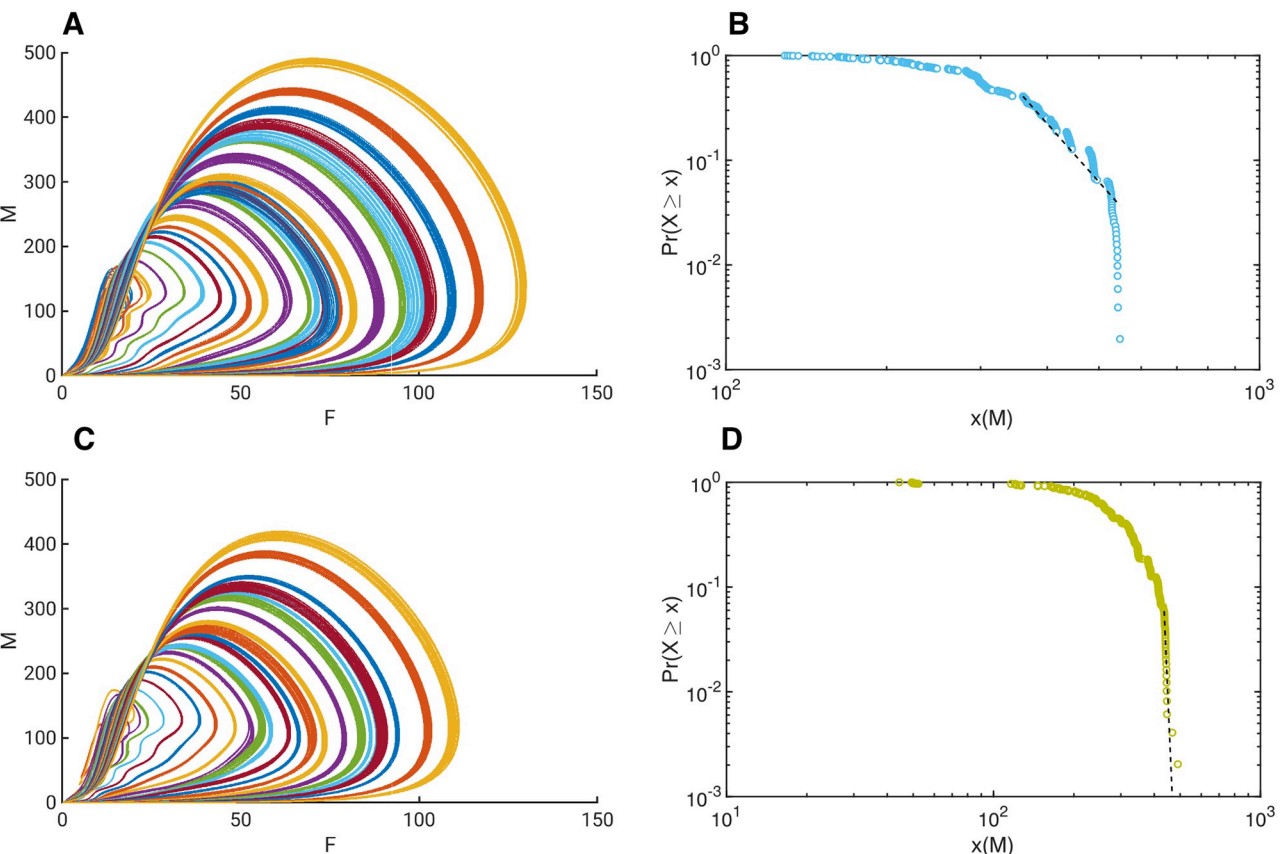

**Fig 5. Global positive or negative feedback causing enhancement or loss of power relations.** *A* Total dynamic behaviour of a network of sixteen coupled rate controlled oscillators with global positive feedback. The feedback pushes the total dynamic behaviour to new heights. Each coloured line is one stable oscillation with positive feedback. *B* Power-law estimation of the maxima of the positive feedback network with the fitted power function (dotted line) superimposed with minimum $x(M) = 360.24$ and slope $\alpha = -6.74$. *C* Total dynamic behaviour of a network of sixteen coupled rate controlled oscillators, differently coloured lines indicate different stable oscillations, with negative global feedback. The negative feedback reduces the behaviour. *D* Power-law estimation of the maxima of the negative feedback network with the fitted power function (dotted line) superimposed with minimum $x(M) = 435.69$ and slope $\alpha = -59.29$.

negative feedback control, the global negative feedback can stabilise individual global states. This in effect counteracts the local connectivity, but at the expense of less dynamic capabilities. Due to the currently available network connectivity the network has become less complex.

## Conclusion

Self-Organised Criticality emerges from local nonlinear interactions of Rate Control of Chaos controlled elements. This allows critical states to exist where the global dynamics, as expressed by the total unweighted sum of each element, is dynamically stable. Multiple states are achievable for the dynamic system, driving the system towards the desired global state by perturbing individual oscillators. Adjusting the local connectivity, in effect recruiting more elements, allows different dynamic states to emerge. Increasing the number of elements allows domains of different relations between perturbations and total behaviour to emerge. These apparent relations, whether power-law, exponential or linear relations, appear due to the networks' self-stabilising properties. Given that he primary aim of any biological system is to maintain stability, the exact nature of the emerging relation becomes therefore a mere epiphenomenon of the ability of the system to stabilise and control its complex dynamics. Furthermore, providing localised global feedback may allow other critical states to also become available, or allow the control of global behaviour into a more limited stable domain.

The described models are clearly in a critical dynamic state, and can readily change state due to local perturbations. This critical state is the result from the RCC controlled systems with local interaction between the units and is therefore self-organising. Lastly, the emerging power law relations and other relations are the result of the nonlinear interactions of the oscillatory units. Therefore, these models are said to describe Self-Organising Criticality because it matches the characteristic key features: non-trivial scaling, spatiotemporal power-law relations (domain bound), and self tuning to the critical state [15].

Biosystems can therefore emerge from the localised interactions between controlled nonlinear systems, creating the perfect combination of complexities that supersedes the limitations of linear systems, avoids the instability of chaotic nonlinear systems, and limits the domain of self-emergent critical systems. This opens up the possibility of innovative research in controlled nonlinear biological dynamics with direct applications to health, engineering control, and human wellbeing.

## Author Contributions

**Conceptualization:** Tjeerd V. olde Scheper.

**Data curation:** Tjeerd V. olde Scheper.

**Formal analysis:** Tjeerd V. olde Scheper.

**Funding acquisition:** Tjeerd V. olde Scheper.

**Investigation:** Tjeerd V. olde Scheper.

**Methodology:** Tjeerd V. olde Scheper.

**Project administration:** Tjeerd V. olde Scheper.

**Resources:** Tjeerd V. olde Scheper.

**Software:** Tjeerd V. olde Scheper.

**Supervision:** Tjeerd V. olde Scheper.

**Validation:** Tjeerd V. olde Scheper.

**Visualization:** Tjeerd V. olde Scheper.

**Writing – original draft:** Tjeerd V. olde Scheper.

**Writing – review & editing:** Tjeerd V. olde Scheper.

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
