## [Decision Letter · Decision Letter 0]

15 Jul 2021

PONE-D-21-16716

Controlled bio-inspired self-organised criticality

PLOS ONE

Dear Dr. olde Scheper,

Thank you for submitting your manuscript to PLOS ONE. After careful consideration, we feel that it has merit but does not fully meet PLOS ONE’s publication criteria as it currently stands. Therefore, we invite you to submit a revised version of the manuscript that addresses the points raised during the review process.

We look forward to receiving your revised manuscript.

Kind regards,

Jun Ma, Dr.

Academic Editor

PLOS ONE

Journal Requirements:

2. Please upload a new copy of Figure 1, 3 and 5 as the detail is not clear. Please follow the link for more information: " ext-link-type="uri" xlink:type="simple">https://blogs.plos.org/plos/2019/06/looking-good-tips-for-creating-your-plos-figures-graphics/"
https://blogs.plos.org/plos/2019/06/looking-good-tips-for-creating-your-plos-figures-graphics/

Reviewer #1: Based on a nonlinear model of a bienzymatic cycle Scheper demonstrated numerically that rate control of chaos (RCC) may be a suitable robust mechanism to produce Self-Organised Criticality (SOC). The theoretical results are beneficial to in-depth understanding of biological control mechanisms. The theme of this paper is very significant and interesting. But there are certainly severe problems which must be solved well before publication.

1. The organizational framework of manuscript is very terrible, especially the section of introduction. The author must rewrite the section to let readers understood the keynotes easily.

2. The related methods used to generate SOC in biosystems should be introduced in brief to get a bird's-eye view of this area

3. Whether the main results obtained here depends the network structure?

4. Can RCC be used in another biological chaos model to generate SOC?

5. Whether the main conclusions can be maintained if noise exists in the model？

6. The author should carefully check the clerical errors and mistakes, such as “Given that he primary”. 

---

## [Author Response · Author response to Decision Letter 0]

27 Sep 2021

Thank you very much for your comments and queries. I have included a letter of response to explain the revisions, which I hope are appropriate and address your concerns. The main issue seemed to be the introduction which has been restructured and revised with additional material added as requested.

---

## [Decision Letter · Decision Letter 1]

2 Nov 2021

Controlled bio-inspired self-organised criticality

PONE-D-21-16716R1

Dear Dr. olde Scheper,

We’re pleased to inform you that your manuscript has been judged scientifically suitable for publication and will be formally accepted for publication once it meets all outstanding technical requirements.

Kind regards,

Jun Ma, Dr.

Academic Editor

PLOS ONE

Reviewer #1: The problems I concerned have been addressed. Therefore, I recommend this paper to be publised in the currect version.

---

## [Editor Report · Acceptance letter]

12 Jan 2022

PONE-D-21-16716R1 

Controlled bio-inspired self-organised criticality 

Dear Dr. olde Scheper:

I'm pleased to inform you that your manuscript has been deemed suitable for publication in PLOS ONE. Congratulations! Your manuscript is now with our production department. 

Kind regards, 

on behalf of

Dr. and Pro. Jun Ma 

Academic Editor

PLOS ONE